# Reduce Human Labor On Evaluating Conversational Information Retrieval System: A Human-Machine Collaboration Approach

**Chen Huang,    Peixin Qin,    Wenqiang Lei[†],    Jiancheng Lv**
College of Computer Science, Sichuan University
{huangc.scu, qinpeixin.scu, wenqianglei}@gmail.com

## Abstract

Evaluating conversational information retrieval (CIR) systems is a challenging task that requires a significant amount of human labor for annotation. It is imperative to invest significant effort into researching more labor-effective methods for evaluating CIR systems. To touch upon this challenge, we take the first step to involve active testing in CIR evaluation and propose a novel method called HumCoE. It strategically selects a few data for human annotation and then calibrates the evaluation results to eliminate evaluation biases. As such, it makes an accurate evaluation of the CIR system at low human labor. We experimentally reveal that it consumes less than 1% of human labor and achieves a consistency rate of 95%-99% with human evaluation results. This emphasizes the superiority of our method.

## 1 Introduction

As evidenced by various studies (Zhang et al., 2018; Aliannejadi et al., 2021), the conversational information retrieval (CIR) system has shown its effectiveness in domains such as conversational question answering (Reddy et al., 2019) and conversational recommendation (Lei et al., 2020b). However, accurate evaluation of CIR systems continues to be a rapidly evolving research area due to the multi-turn nature of user-system conversations (Zamani et al., 2022; Sekulić et al., 2022; Gao et al., 2021). It often requires the user to constantly interact with the CIR system and then evaluate the system based on its real-time responses, which are labor-expensive and generally only feasible in an industry lab (Labhishetty and Zhai, 2021).

To alleviate this problem, current practices prepare a set of input data and employ humans to either manually annotate the answer to be compared with the system-generated results or directly score the corresponding system output (Larson et al., 2019;

Li et al., 2022). For example, Braun et al. (2017); Choi et al. (2018) require the human to answer the questions in a pre-collected question set and use these answers as ground truth to evaluate the model-generated results. Li et al. (2022) requires the human to validate and re-annotate the pre-collected conversation dataset to provide finer-grained labels for the input and characterize the system performance based on such labels. Moreover, Aliannejadi et al. (2020) employs the human to directly rate each response of the system on relevance and naturalness. While these methods show their effectiveness in CIR evaluation, they still require traversing the entire dataset, resulting in significant human labor costs (Budzianowski et al., 2018; Dalton et al., 2020; Adlakha et al., 2022). Consequently, it is imperative to invest significant effort into researching more labor-effective methods for evaluating CIR systems.

To this end, we take the first step to involve *active testing*[1] for the label-effective CIR evaluation, proposing a novel method called Human-machine Collaborative Evaluation (*HumCoE*, for short). The idea is to strategically select a few data for the human to annotate the system outputs (e.g., provide an answer to the question), and then calibrate the evaluation results to eliminate evaluation biases. Specifically, given an affordable number of human interactions, HumCoE starts by selecting a subset of representative data and requesting human annotation or scoring. In this process, the representative data is measured based on the probability of inconsistency between CIR's prediction and a pseudo-label that is generated by a surrogate machine. After that, HumCoE assigns an importance weight to each selected data and calibrates the evaluation results. As such, the sampling bias is eliminated, and hence the evaluation accuracy is

---

[†] Correspondence to Wenqiang Lei.

[1]It estimates model performance (i.e., CIR system, in our case) by selecting a small subset of data for human annotation (i.e., evaluation)(Rahman et al., 2020; Nguyen et al., 2018).

largely improved.

We demonstrate the effectiveness of HumCoE on typical CIR tasks, including asking clarifying questions and the conversational question answering. In our experiments, we assess the task success rate and quality of system responses using automatic evaluation metrics on pre-collected conversational datasets. Additionally, we employ the human to assess the relevance and naturalness scores of system responses. Our experiments indicate that HumCoE produces results that are extremely comparable to those of a human-based evaluation yet require very little labor. It consumes less than 1% of human labor and achieves a consistency rate of 95%-99% with human evaluation results on all tasks. Compared to other classical active testing methods, our approach achieves a higher evaluation consistency rate by an average of 11.75%. We also explore our method in the context of ChatGPT. We notice that ChatGPT does not provide satisfactory evaluation consistency with human evaluation, with an average consistency rate of 88.24%. But, after introducing a small number of labor costs (0.68%), HumCoE improves the evaluation consistency by 9.59%. This highlights the importance of human-machine collaboration for efficient CIR evaluation.

Evaluating any highly interactive process is challenging. In this paper, we experimentally reveal our effectiveness and efficiency on the CIR evaluation task. We believe that our work could provide a landmark for future human-machine-based CIR evaluation research. To sum up, we claim the following contributions.

- We call attention to the challenge of designing labor-effective methods for accurately evaluating CIR systems.

- For the first time, we introduce active testing to evaluate the CIR system and propose a novel method called HumCoE. It strategically selects representative data for human annotation and carefully calibrates the evaluation results.

- We verify our effectiveness with empirical studies, which demonstrate that our method enables accurate evaluation with minimal human effort. We set a landmark for future human-machine-based CIR evaluation research.

## 2 Related Works

As our focus is on utilizing both human and machine in evaluating the CIR system via active test-

ing, we offer literature reviews on CIR evaluation and active testing. For more discussion on human-machine evaluation, please refer to Appendix E.

**CIR Evaluation.** Given an evaluation metric, existing methods use user utterances and the system's responses to evaluate the CIR system. Such information is either obtained from real-time user-system conversations during online evaluation or contained in a pre-collected conversational dataset during offline evaluation (Zamani et al., 2022). Online evaluation (Zamani et al., 2022; Aliannejadi et al., 2019; Li et al., 2019) employs real users to interact with the CIR system several times and dynamically obtains user utterances and the system's responses for evaluation (Ram et al., 2018; Park et al., 2020). However, a large number of interactions require a huge labor cost, which limits the feasibility and application scenarios of this method. Alternatively, the offline evaluation method is economically inexpensive as it permits evaluations via static conversational datasets (Yang et al., 2018; Budzianowski et al., 2018; Henderson et al., 2019). During the evaluation, the CIR system only receives rigid and fixed questions or responses, extracted from the static conversational datasets. However, let alone that such methods do not transfer well to the real-world scenario and often lead to unnatural conversations (Lipani et al., 2021; Sekulić et al., 2022), constructing the dataset still requires a significant amount of human labor (Budzianowski et al., 2018; Dalton et al., 2020; Adlakha et al., 2022), especially when evaluating specific features of the system that are absent from its training data (Braun et al., 2017; Larson et al., 2019; Li et al., 2022). As a result, how to design labor-effective methods for CIR evaluation is still an unaddressed problem. In this paper, we propose an evaluation method based on human-machine collaboration. We experimentally show that it makes an accurate evaluation that is extremely comparable to those of the human-based evaluation with low human labor.

**Active Testing**. Different from active learning, which targets selecting partial data to train a better model, *active testing* focuses on reliably estimating model performance by selecting less unlabeled testing data for human annotation (Nguyen et al., 2018; Li et al., 2020). By allocating data between humans and machines, active testing methods achieve collaboration between humans and machines. Thus, data selection plays an important role in active test-

ing. Usually, hard-to-predict data with large model prediction uncertainties (Rahman et al., 2020) or inconsistencies in model predictions (Nguyen et al., 2018) are always preferred. More broadly, Zhang et al. (2021a) has shown its evaluation effectiveness on the task of malevolence identification. It assigns dialogues with high model uncertainty to the human while assigning the rest to the machine for evaluation. Inspired by the idea of active testing, we take the first step to apply it to CIR evaluation by selecting partial hard data for human evaluation and carefully calibrating the results. According to our experiments, although human evaluation costs are reduced, existing methods (Zhang et al., 2021a; Rahman et al., 2020; Nguyen et al., 2018) still suffer low evaluation accuracy because the machine may be error-prone and the evaluation results are uncalibrated. This emphasizes the importance of evaluation calibration.

## 3 Task Definition

Given an evaluation metric $EM$ and a test conversational dataset[2] $D$, the evaluation $\tau$ is completed by calculating the metric value on the given dataset, described by the following equations.

$$\tau(D, EM, Y_D) = \frac{\sum_{d_i \in D, y_i \in Y_D} EM(d_i, y_i)}{|D|}. \tag{1}$$

Here, $d_i = \{x_i, u_i\}$ represents the $i$-th user-CIR interaction that consists of multiple conversational turns. Specifically, $x_i = (x_i^1, x_i^2, ..., x_i^k)$ is a list of responses of CIR $f$, where $x_i^j$ is $j$-th CIR response in $i$-th user-CIR interaction. Similarly, $u_i = (u_i^1, u_i^2, ..., u_i^k)$ is a list of user responses, and $y_i$ is a list of user-annotated ground truth CIR responses. Take the conversational question answer as an example (Peng et al., 2022), $x_i^j$ refers to the system returned answer to $j$-th user query $u_i^j$ in $i$-th user-CIR interaction. In this case, $EM$ could refer to the Rouge-L to evaluate the word overlap between the CIR response $x_i^j$ and ground truth $y_i^j$, i.e., $EM(d_i, y_i) = 1/k \sum_j \text{Rouge-L}(x_i^j, y_i^j)$. $EM$ could also refer to the Accuracy to evaluate the task success rate of CIR retrieving the correct answer, i.e., $EM(d_i, y_i) = 1/k \sum_j \mathbb{I}(x_i^j = y_i^j)$, where $\mathbb{I}(\cdot)$ is an indicator. We denote $\tau(D, EM, Y_D)$ as human evaluation, where the size of $D$ is usually large to guarantee the statistical significance of the evaluation results.

---

[2]Usually, It is a pre-collected dataset annotated by humans. Ours are created automatically (cf. section 4).

## 4 Methodology

There are two steps in HumCoE: 1) It starts by selecting a subset of representative data $D_s$ and requesting human annotation or scoring $Y_{D_s}$, where the representative data $D_s$ are selected based on a surrogate machine. 2) After that, it assigns an importance weight to each selected data and calibrates the evaluation results.

**Step 1: Surrogate Machine based Data Sampling.** We aim to select representative data for human annotation. In this process, the representative data is measured based on the probability of inconsistency between CIR's prediction and a pseudo-label that is generated by a surrogate machine.

Before human annotation, we involve a surrogate machine $g$ as a user proxy to generate pseudo-labels. Taking the conversational question-answering task as an example, the surrogate machine is required to answer the input questions. The output answers are stored as pseudo-labels. Formally, given the surrogate machine-annotated dataset $D_1$, we denote $d_i = \{x_i, v_i\} \in D_1$ and $v_i$ is a list of proxy responses. Usually, a surrogate machine $g$ is trained or fine-tuned based on the same training data as $f$. Meanwhile, designing $g$ with a different architecture from $f$ is always preferred to encourage response diversity between $f$ and $g$. For example, one could set GPT-2 as $f$ and BART as $g$ to generate responses during the conversations.

Borrowing the idea of active testing methods (Zhang et al., 2021a), we aim to select a subset of representative data $D_s$ that are hard for the CIR system to properly handle for human annotation $Y_{D_s}$. The "hardness" in this paper is measured by the probability of the CIR system making mistakes on pseudo-labels $\hat{Y}_{D_1}$, rather than the model uncertainty (Zhang et al., 2021a). Namely, we prefer data with a high probability of inconsistency between CIR's predictions and pseudo labels. Technically, we measure the hardness as follows.

$$q(d_i) \propto 1 - EM(d_i, \hat{y}_i). \tag{2}$$

Large $q(d_i)$ means that the performance of CIR $f$ on $d_i$ in terms of a metric $EM$ may not be satisfying. Finally, given the maximum number of human interactions $T$, we sample a subset $D_s$ with the size $|D_s| = T$ according to probability $q$. The selected data are assigned to the human for evaluation and obtaining the ground truth labels.

**Step 2: Evaluation Calibration.** Using hard samples to evaluate a CIR system inevitably leads to biased evaluation results. It tends to underestimate the effectiveness of the system. To this end, HumCoE assigns a weight to each sample and corrects its importance in the evaluation results, thereby moderating the negative impact of hard samples on the evaluation results. Formally, we follow previous work on active learning (Farquhar et al., 2021) and design the weight $w_i$ of sample $d_i$ as follows.

$$w_i = 1 + \frac{N-T}{N-1}\left(\frac{1}{Nq(d_i)} - 1\right), \quad (3)$$

where $N$ is the size of the proxy-CIR conversation dataset $D_1$. In this case, we assign smaller weights to data with large $q(d_i)$ and penalize the importance of hard data. Different from existing work (Zhang et al., 2021a), we finally calibrate and yield the evaluation result as follows.

$$\hat{\tau}(D_s, EM, Y_{D_s}) = \frac{\sum_{d_i \in D_s, y_i \in Y_{D_s}} w_i EM(d_i, y_i)}{|D_s|}. \quad (4)$$

## 5 Experiments

Our method formulates a human-machine collaboration to reduce the labor of human evaluation while retaining its evaluation accuracy. Therefore, the main experiments in section 5.1 are designed to assess if HumCoE and active testing based evaluation methods could approximate the results of the human evaluation at a very few cost on human labor. Finally, we further perform ablation studies on HumCoE in section 5.2 to analyze its properties.

**Baselines.** We utilize human evaluation to evaluate a given CIR system with no limitations on human labor costs. We compare our method to other active testing methods that are most relevant to us, including HMCEval (Zhang et al., 2021a), Active Top-K (Rahman et al., 2020), and Most-Confident Mistake (MCM) (Nguyen et al., 2018).

- **HMCEval** (Zhang et al., 2021a). It is intended for assessing malevolence in dialogues, and it frames dialogue evaluation as a sample assignment problem. To achieve this, HMCEval presents a constrained optimization problem.

- **Active Top-K** (Rahman et al., 2020) draws the top-k hard samples from the CIR's ranked uncertainty scores.

- **MCM** (Nguyen et al., 2018) selects hard samples whose CIR prediction confidences are very high, and whose CIR prediction results are inconsistent with those of the surrogate machine. For example, if the CIR predicts a positive label with high probability while the surrogate machine assigns a negative pseudo label, it suggests that the surrogate machine's label may be incorrect and needs human annotation for correction.

These methods select hard samples for human evaluation like us but adopt different sampling strategies. Note that we take the first step to utilize active testing to evaluate CIR. In our experiments, we re-implement other active testing methods to fit in the CIR scenario.

**Implementation Details**. Considering evaluating any highly interactive process is challenging, we follow previous works (Hwa, 2000; Kristjansson et al., 2004) and assume the time spent by the human evaluations on each sample $d_i$ is constant[3]. Therefore, we use the number of human evaluations to estimate the human labor, i.e., the size of $D_s$. In our experiments, we limit the size of $D_s$ in $\{5, 10, 15, 20, 25, 30\}$ to verify the effectiveness of different evaluation methods with minimal labor costs. Considering the randomness of data sampling, we run the evaluation 100 times and report the mean estimation results $\overline{\tau}$.

$$\overline{\tau} = \frac{\sum_{s_i \in Seed} \hat{\tau}(D_s, EM, Y_{D_s}|s_i)}{|Seed|} \quad (5)$$

Each time, the random seed is fixed and shared with different evaluation method. For implementation details, we refer readers to Appendix A.

**Evaluation Metrics.** We assess if HumCoE could approximate the results of the human evaluation at a very few cost on human interactions. Therefore, we consider the following metrics.

- Evaluation Consistency. We calculate the absolute value of the inconsistency between human evaluation $\tau(D, EM, Y_D)$ and a given evaluation method $\tau_0$, i.e., $\Delta\tau = |\tau(D, EM, Y_D) - \tau_0|$. The lower the inconsistency, the greater the evaluation accuracy of the given evaluation method. We further report the consistency ratio as $\tau^\circ = 1 - \frac{\Delta\tau}{\tau(D,EM)}$.

---

[3]Treating human labor equally on each data point is widely used and necessary in current research (Rahman et al., 2020; Aliannejadi and Trippas, 2022; Desmond et al., 2021).

- **Evaluation Stability.** It measures the stability of HumCoE since the data sampling introduces randomness to a certain extent. We consider the variance $\tau_v$ of multiple evaluation results and the squared error $\tau_e$ to highlight the evaluation stability in a finer-grained way.

$$\tau_v = \frac{\sum_{s_i \in Seed} (\hat{\tau}(D_s, EM, Y_{D_s}|s_i) - \overline{\tau})^2}{|Seed|}$$
$$\tau_e = \frac{\sum_{s_i \in Seed} (\hat{\tau}(D_s, EM, Y_{D_s}|s_i) - \tau(D, EM, Y_D))^2}{|Seed|}$$
$$(6)$$

## 5.1 Main Results

We verify if HumCoE and other active testing methods approximate the results of the human evaluation at a very few cost on human labor. We show HumCoE's effectiveness on typical CIR tasks, including the conversational question answer and clarifying question, suggested by a recent survey on CIR (Gao et al., 2022). We evaluate the task success rate as well as the quality of system responses. Regarding clarifying questions, we consider two settings: the question is retrieved from pre-defined candidates or generated by a language model. We leave the implementation details in Appendix A.

### 5.1.1 Task 1: Conversational QA

**Task Description.** Question answering is concerned with delivering relatively short pieces of information to answer the queries. Conversational question-answering systems need to handle the more complex linguistic characteristics of conversations, anaphoras or ellipses may occur in question sentences (Vakulenko et al., 2021). In the user-CIR conversations, the CIR system must respond to every question posed by the user.

**Datasets & Metrics.** Following previous works (Kim et al., 2022; Peng et al., 2022), we use the CoQA dataset (Reddy et al., 2019). CoQA is essentially a collection of dialogues of questions and answers from crowded human annotations. Following (Peng et al., 2022; Bao et al., 2022), Rouge-L is used to assess the word overlap between the prediction and the ground truth answer.

**Task Model in CIR.** As the task model employed in CIR, we use the recently proposed and open-sourced method GODEL (Peng et al., 2022) and evaluate the effectiveness of GODEL on CoQA. In our experiments, both GODEL-base and GODEL-large are evaluated on the test data, which consists of 7979 conversations. In HumCoE, GODEL-base and GODEL-large are used

interchangeably as the surrogate machines. In particular, we use GODEL-base as the task model $f$ and GODEL-large as the surrogate machine $g$, and vice versa. The predicted answers of the surrogate machine are utilized in Eq. 2 as pseudo-labels, while Rouge-L plays the role of $EM$. Since CoQA are fully annotated by humans, human evaluation can be easily and accurately computed. Based on the test data, we compared the answers generated by the GODEL-base and GODEL-large on the test data with the manually annotated answers, respectively. Human evaluation of the two models yields results of 68.56 and 75.09.

**Evaluation Results.** As shown in Table 1, HumCoE consumes only less than 0.38% (i.e., 30 out of 7979) of human labor and achieves an average consistency rate of 99.41% with human evaluation, which verifies the effectiveness and efficiency of our method on the conversational question answering task. We also found that the size of the parameters of the surrogate machine $g$ and task model $f$ has little impact on the evaluation results of HumCoE. Meanwhile, compared to other active testing methods, our method achieves an average evaluation accuracy of 99.17% and 99.64% on two task models for different labor costs. This translates into 46.86% and 36.51% performance gains over Active Top-K, 13.23% and 3.81% performance gains over HMCEval, and 13.37% and 4.04% performance gains over MCM.

Table 1: Evaluation Consistency on CoQA. We report the consistency ratio $\tau^\circ$ (%) and the inconsistency $\Delta\tau$. '#' means the number of human interactions, and the number in the table is in the form of $\tau^\circ_{(\Delta\tau)}$.

| CIR $f$ | # | HumCoE (ours) | Active Top-K | HMCEval | MCM |
|---|---|---|---|---|---|
| Godel-base | 5 | $99.69_{(0.21)}$ | $56.96_{(29.51)}$ | $85.91_{(9.66)}$ | $85.92_{(9.65)}$ |
| | 10 | $98.73_{(0.87)}$ | $60.08_{(27.37)}$ | $85.91_{(9.66)}$ | $85.91_{(9.66)}$ |
| | 15 | $98.44_{(1.07)}$ | $40.05_{(41.10)}$ | $85.91_{(9.66)}$ | $85.91_{(9.66)}$ |
| | 20 | $99.58_{(0.29)}$ | $51.93_{(32.96)}$ | $85.94_{(9.64)}$ | $85.91_{(9.66)}$ |
| | 25 | $99.20_{(0.55)}$ | $53.21_{(32.08)}$ | $85.98_{(9.61)}$ | $85.90_{(9.67)}$ |
| | 30 | $99.39_{(0.42)}$ | $51.63_{(33.16)}$ | $85.97_{(9.62)}$ | $85.87_{(9.69)}$ |
| Godel-large | 5 | $99.84_{(0.12)}$ | $53.27_{(35.09)}$ | $95.82_{(3.14)}$ | $95.78_{(3.17)}$ |
| | 10 | $99.28_{(0.54)}$ | $53.27_{(35.09)}$ | $95.82_{(3.14)}$ | $95.71_{(3.22)}$ |
| | 15 | $99.72_{(0.21)}$ | $74.58_{(19.09)}$ | $95.79_{(3.16)}$ | $95.63_{(3.28)}$ |
| | 20 | $99.33_{(0.50)}$ | $72.58_{(20.59)}$ | $95.81_{(3.15)}$ | $95.54_{(3.35)}$ |
| | 25 | $99.91_{(0.07)}$ | $63.39_{(27.49)}$ | $95.86_{(3.11)}$ | $95.50_{(3.38)}$ |
| | 30 | $99.75_{(0.19)}$ | $61.70_{(28.76)}$ | $95.86_{(3.11)}$ | $95.46_{(3.41)}$ |
| **Avg. $\tau^\circ$ (%)** | | **99.41** | 57.72 | 90.88 | 90.75 |

### 5.1.2 Task 2: Clarifying Question (Retrieved)

**Task Description.** With a mixed-initiative conversational search, the CIR can take the lead and ask the user clarifying questions to clear up the ambiguity in the user's query (Sekulić et al., 2022; Keyvan

and Huang, 2022). Formally, the user expresses their demand in the initial query form, which is subsequently provided to the CIR system. The user needs to generate an answer to the system's clarifying question. The CIR system aims to elucidate the information need through a series of clarifying questions and return the correct answer.

In accordance with earlier studies (Aliannejadi et al., 2020), we assume that the clarifying questions are retrieved from a pre-defined question pool (see section 5.1.3 for model-generated clarifying questions). In this task, we evaluate the task success rate (i.e., if it returns the correct answer at the end), and ignore the evaluation on question quality as the question is pre-defined[4].

**Datasets & Metrics.** ClariQ (Aliannejadi et al., 2021, 2020) is employed to evaluate our methods. Given a conversation with multiple turns, we evaluate if the CIR could select a candidate question that would best clarify the user's intent. Thus, it is a classification problem, and we utilize the accuracy metric to measure the task success rate.

**Task Model in CIR.** Following Aliannejadi et al. (2021), we use BERT-base as CIR model $f$. As for the surrogate machine $g$, we consider ERNIE-base for simplicity. Notably, the two models are also used interchangeably as $f$ and $g$. The retrieved questions of the surrogate machine are utilized in Eq. 2 as pseudo-labels, and the accuracy metric serves as $EM$. Later, these two models are fine-tuned on ClariQ training data and then evaluated on the test data, which contains 4411 conversations. Similarly, since ClariQ is fully human-annotated, the results of human evaluation on the BERT and ERNIE are 88.03% and 91.79%, respectively.

Table 2: Evaluation Consistency on ClariQ, where the clarifying questions are retrieved.

| CIR $f$ | # | HumCoE (ours) | Active Top-K | HMCEval | MCM |
|---|---|---|---|---|---|
| BERT | 5 | **99.76**$_{(0.21)}$ | 90.88$_{(8.03)}$ | 97.52$_{(2.18)}$ | 97.52$_{(2.18)}$ |
| | 10 | **99.69**$_{(0.27)}$ | 56.80$_{(38.03)}$ | 97.52$_{(2.18)}$ | 97.52$_{(2.18)}$ |
| | 15 | **99.93**$_{(0.06)}$ | 53.02$_{(41.36)}$ | 97.50$_{(2.20)}$ | 97.50$_{(2.20)}$ |
| | 20 | **99.91**$_{(0.08)}$ | 51.12$_{(43.03)}$ | 97.40$_{(2.29)}$ | 97.40$_{(2.29)}$ |
| | 25 | **99.75**$_{(0.22)}$ | 59.07$_{(36.03)}$ | 97.32$_{(2.36)}$ | 97.32$_{(2.36)}$ |
| | 30 | **99.73**$_{(0.24)}$ | 60.58$_{(34.70)}$ | 97.32$_{(2.36)}$ | 97.30$_{(2.38)}$ |
| ERNIE | 5 | 97.01$_{(2.74)}$ | 43.58$_{(51.79)}$ | **98.25**$_{(1.61)}$ | **98.25**$_{(1.61)}$ |
| | 10 | **99.74**$_{(0.24)}$ | 76.26$_{(21.79)}$ | 98.34$_{(1.52)}$ | 98.34$_{(1.52)}$ |
| | 15 | **99.56**$_{(0.40)}$ | 79.89$_{(18.46)}$ | 98.42$_{(1.45)}$ | 98.42$_{(1.45)}$ |
| | 20 | **99.87**$_{(0.12)}$ | 76.26$_{(21.79)}$ | 98.40$_{(1.47)}$ | 98.50$_{(1.38)}$ |
| | 25 | **99.67**$_{(0.30)}$ | 82.80$_{(15.79)}$ | 98.42$_{(1.45)}$ | 98.52$_{(1.36)}$ |
| | 30 | **99.74**$_{(0.24)}$ | 76.26$_{(21.79)}$ | 98.47$_{(1.40)}$ | 98.59$_{(1.29)}$ |
| **Avg. $\tau^\circ$ (%)** | | **99.53** | 67.21 | 97.91 | 97.37 |

[4]We also ignore the interaction efficiency as it could be easily computed without human involvement.

**Evaluation Results.** Table 2 demonstrates how our HumCoE continues to achieve promising results and improve over the baseline on the clarifying question. In particular, our HumCoE consumes only less than 0.68% of human labor (i.e., 30 out of 4411) and achieves an average consistency rate of 99.53% with human evaluation. It achieves evaluation accuracy of 99.80% and 99.27% on two task models with different labor costs. Compared to other active testing methods, HumCoE enjoys 32.32% performance gains over Active Top-K (i.e., 37.89% on BERT and 26.76% on ERNIE), 1.62% performance gains over HMCEval (i.e., 2.37% on BERT and 0.89% on ERNIE), and 2.16% performance gains over MCM (i.e., 2.37% on BERT and 0.83% on ERNIE).

### 5.1.3 Task 3: Clarifying Question (Generated)

**Task Description.** The CIR system that *generates* clarifying questions is evaluated in this task. Here, we switch our attention to assessing the quality of the model-generated questions.

**Datasets & Metrics.** Following earlier studies (Aliannejadi et al., 2021, 2020), we also consider ClariQ. To evaluate the quality of the generated questions, we consider two types of evaluation metrics, i.e., automatic and user-based metrics. Specifically, Rouge-L is used as the automatic evaluation metric, where human-annotated questions act as the references. Regarding the user-based evaluation, we follow previous work (Aliannejadi et al., 2020) and employ users to assess the relevance and naturalness of each system response. On the one hand, the relevance score (0-5) measures if the clarifying question is relevant to the user's information needs. The naturalness score (0–5), on the other hand, measures if the clarifying question is natural in the context of the conversation.

**Task Model in CIR.** Following (Sekulić et al., 2021), we alternate between using BART-base and GPT2-base as the task model $f$ and surrogate machine $g$. These two models are fine-tuned on ClariQ training data. Here, we report the results of human evaluation on the BART and GPT-2: the Rouge-L score on BART and GPT-2 are 37.57 and 48.46, respectively. Meanwhile, the relevance score and naturalness scores for GPT-2 are 3.09 and 3.51, while the scores for BART are 3.82 and 3.85, respectively.

**Automatic calculation on relevance and naturalness scores.** The process of manual scoring is difficult to explicitly write into a specific for-

Table 3: Question quality evaluation on ClariQ, where the questions are model generated. We report the consistency ratio $\tau^\circ$ (%) and inconsistency $\Delta\tau$. For consistency rate of ChatGPT, it achieves 89.32% on relevance score and 89.46% on naturalness score when $f$=GPT-2. For BART, ChatGPT achieves 88.22% and 85.97% on those scores. After introducing a small number of labor cost, HumCoE further improves the evaluation consistency.

| CIR $f$ | # | HumCoE (ours) | Active Top-K | HMCEval | MCM | CIR $f$ | # | HumCoE (ours) | Active Top-K | HMCEval | MCM |
|---|---|---|---|---|---|---|---|---|---|---|---|
| | | | | | **Clarifying Question (Generated) with Offline Automatic Metric** | | | | | | |
| GPT-2 (Rouge-L) | 5 | **99.57**$_{(0.16)}$ | 89.97$_{(3.77)}$ | 94.01$_{(2.25)}$ | 94.14$_{(2.20)}$ | BART (Rouge-L) | 5 | **93.56**$_{(3.12)}$ | 69.93$_{(14.57)}$ | 82.07$_{(8.69)}$ | 82.81$_{(8.33)}$ |
| | 10 | **98.86**$_{(0.43)}$ | 81.16$_{(7.08)}$ | 94.01$_{(2.25)}$ | 93.88$_{(2.30)}$ | | 10 | **94.99**$_{(2.43)}$ | 69.58$_{(14.74)}$ | 82.15$_{(8.65)}$ | 83.41$_{(8.04)}$ |
| | 15 | **97.55**$_{(0.92)}$ | 88.32$_{(4.39)}$ | 93.85$_{(2.31)}$ | 93.77$_{(2.34)}$ | | 15 | **95.23**$_{(2.31)}$ | 71.42$_{(13.85)}$ | 82.83$_{(8.32)}$ | 84.38$_{(7.57)}$ |
| | 20 | **96.75**$_{(1.22)}$ | 79.82$_{(7.58)}$ | 93.59$_{(2.41)}$ | 93.05$_{(2.61)}$ | | 20 | **96.10**$_{(1.89)}$ | 67.79$_{(15.61)}$ | 83.04$_{(8.22)}$ | 85.27$_{(7.14)}$ |
| | 25 | **95.26**$_{(1.78)}$ | 81.29$_{(7.03)}$ | 93.80$_{(2.33)}$ | 92.76$_{(2.72)}$ | | 25 | **95.60**$_{(2.13)}$ | 72.29$_{(13.43)}$ | 83.45$_{(8.02)}$ | 85.62$_{(6.97)}$ |
| | 30 | 93.43$_{(2.47)}$ | 85.01$_{(5.63)}$ | **93.69**$_{(2.37)}$ | 92.44$_{(2.84)}$ | | 30 | **94.82**$_{(2.51)}$ | 76.25$_{(11.51)}$ | 83.55$_{(7.97)}$ | 86.03$_{(6.77)}$ |
| **Avg. $\tau^\circ$ (%)** | | **96.90** | 84.26 | 93.83 | 93.34 | **Avg. $\tau^\circ$ (%)** | | **95.05** | 71.21 | 82.85 | 84.59 |
| | | | | | **Clarifying Question (Generated) with Online Human-based Metrics** | | | | | | |
| GPT-2 (Relevance) | 5 | **99.06**$_{(0.03)}$ | 96.55$_{(0.11)}$ | **99.06**$_{(0.03)}$ | 98.81$_{(0.04)}$ | GPT-2 (Naturalness) | 5 | **98.74**$_{(0.04)}$ | 87.39$_{(0.44)}$ | 89.54$_{(0.37)}$ | 90.03$_{(0.35)}$ |
| | 10 | 98.86$_{(0.04)}$ | 80.82$_{(0.59)}$ | **99.21**$_{(0.02)}$ | 98.38$_{(0.05)}$ | | 10 | **99.62**$_{(0.01)}$ | 85.49$_{(0.51)}$ | 89.79$_{(0.36)}$ | 90.77$_{(0.32)}$ |
| | 15 | 97.74$_{(0.07)}$ | 82.61$_{(0.54)}$ | **98.96**$_{(0.03)}$ | 97.69$_{(0.07)}$ | | 15 | **98.39**$_{(0.06)}$ | 80.43$_{(0.69)}$ | 89.87$_{(0.36)}$ | 91.62$_{(0.29)}$ |
| | 20 | 97.34$_{(0.08)}$ | 78.13$_{(0.68)}$ | **99.04**$_{(0.03)}$ | 96.88$_{(0.10)}$ | | 20 | **98.03**$_{(0.07)}$ | 78.37$_{(0.76)}$ | 89.99$_{(0.35)}$ | 92.29$_{(0.27)}$ |
| | 25 | 97.63$_{(0.07)}$ | 86.21$_{(0.43)}$ | **98.86**$_{(0.04)}$ | 96.20$_{(0.12)}$ | | 25 | **98.18**$_{(0.06)}$ | 80.55$_{(0.68)}$ | 89.96$_{(0.35)}$ | 93.00$_{(0.25)}$ |
| | 30 | 96.81$_{(0.10)}$ | 86.57$_{(0.42)}$ | **98.73**$_{(0.04)}$ | 95.61$_{(0.14)}$ | | 30 | **98.00**$_{(0.07)}$ | 79.79$_{(0.71)}$ | 89.99$_{(0.35)}$ | 93.74$_{(0.22)}$ |
| BART (Relevance) | 5 | **98.86**$_{(0.04)}$ | 94.15$_{(0.22)}$ | 88.55$_{(0.44)}$ | 88.02$_{(0.46)}$ | BART (Naturalness) | 5 | **99.75**$_{(0.01)}$ | 89.97$_{(0.39)}$ | 86.24$_{(0.53)}$ | 85.69$_{(0.55)}$ |
| | 10 | **98.38**$_{(0.06)}$ | 97.64$_{(0.09)}$ | 88.33$_{(0.45)}$ | 87.94$_{(0.46)}$ | | 10 | **98.69**$_{(0.05)}$ | 95.16$_{(0.19)}$ | 86.08$_{(0.54)}$ | 85.51$_{(0.56)}$ |
| | 15 | **97.99**$_{(0.08)}$ | 95.32$_{(0.18)}$ | 88.47$_{(0.44)}$ | 87.45$_{(0.48)}$ | | 15 | **99.16**$_{(0.03)}$ | 95.73$_{(0.16)}$ | 86.36$_{(0.54)}$ | 85.49$_{(0.56)}$ |
| | 20 | **98.19**$_{(0.07)}$ | 97.21$_{(0.11)}$ | 88.57$_{(0.44)}$ | 87.24$_{(0.49)}$ | | 20 | 97.88$_{(0.08)}$ | 96.45$_{(0.14)}$ | 86.42$_{(0.52)}$ | 85.14$_{(0.57)}$ |
| | 25 | **98.03**$_{(0.08)}$ | 93.81$_{(0.24)}$ | 88.59$_{(0.44)}$ | 86.97$_{(0.50)}$ | | 25 | **98.23**$_{(0.07)}$ | 97.23$_{(0.11)}$ | 86.75$_{(0.51)}$ | 84.94$_{(0.58)}$ |
| | 30 | **98.00**$_{(0.08)}$ | 94.44$_{(0.21)}$ | 88.82$_{(0.43)}$ | 86.73$_{(0.51)}$ | | 30 | **98.50**$_{(0.06)}$ | 96.60$_{(0.13)}$ | 86.87$_{(0.51)}$ | 84.65$_{(0.59)}$ |
| **Avg. $\tau^\circ$ (%)** | | **98.07** | 90.29 | 93.77 | 92.33 | **Avg. $\tau^\circ$ (%)** | | **98.60** | 88.60 | 88.16 | 88.57 |

mula. Namely, there is no pre-defined and specific formula like Rouge-L that takes a generated sentence as input and directly outputs the relevance and naturalness scores. Therefore, it is challenging to calculate Eq.2 for relevance and naturalness score evaluation as the $EM$ is unknown. To this end, we design prompts for ChatGPT scoring and achieve automatic calculation of relevance and naturalness scores (see Appendix B for details). In this case, $q(d_i) \propto 1 - \text{ChatGPT}(d_i, \hat{y}_i)$.

Given the results of ChatGPT on GPT-2, the relevance and naturalness scores are 3.42 and 3.14, which translate into an evaluation consistency of 89.32% and 89.46% with human evaluation. We also examine the inter-annotator reliability of the ChatGPT scores in a finer-grained way. In particular, we resort to Krippendorff's alpha (Krippendorff), which is a commonly used reliability coefficient developed to measure the agreement among different annotators. Given human evaluation, ChatGPT scores only achieve 0.321 on relevance and 0.284 on naturalness, implying the agreement between human evaluation and ChatGPT is relatively low. Similarly, regarding the BART, the relevance and naturalness scores from ChatGPT are 3.37 and 3.31, respectively. In this case, ChatGPT scores achieve 88.22% and 85.97% in terms of evaluation accuracy, 0.405 on relevance, and 0.339 on naturalness in terms of Krippendorff's alpha. We highlight that the consistency between

ChatGPT and human evaluation results is not satisfying (on average, 88.24%), and ChatGPT may be difficult to replace humans in achieving fully automated CIR evaluation. Interestingly, by introducing a small number of labor costs (0.68% of human labor), HumCoE corrects the evaluation results of ChatGPT and improves consistency with human evaluation results by 9.59% (see the Evaluation Results below).

**Evaluation Results.** Table 3 summarizes the results of different evaluation methods on different evaluation metrics. Our method consumes only less than 0.68% of human labor (i.e., 30 out of 4411) and achieves an average consistency rate of 95.98% on Rouge-L with human evaluation (i.e., 96.90% on GPT-2 and 95.05% on BART), 98.07% on Relevant score (i.e., 97.88% on GPT-2 and 98.24% on BART), and 98.60% on Naturalness score (i.e., 98.49% on GPT-2 and 98.70% on BART). By introducing a small number of labor costs (0.68% of human labor), HumCoE corrects the evaluation results of ChatGPT and improves the consistency in terms of Relevant score (i.e., 8.56% improvement on GPT-2 and 10.02% on BART) and Naturalness score (i.e., 9.03% improvement on GPT-2 and 12.73% on BART). This motivates the need of human-machine collaboration for CIR evaluation. Meanwhile, we observed that HumCoE's advantage on GPT-2's averaged relevance score is slightly reduced by 1.07% compared to HMCEval.

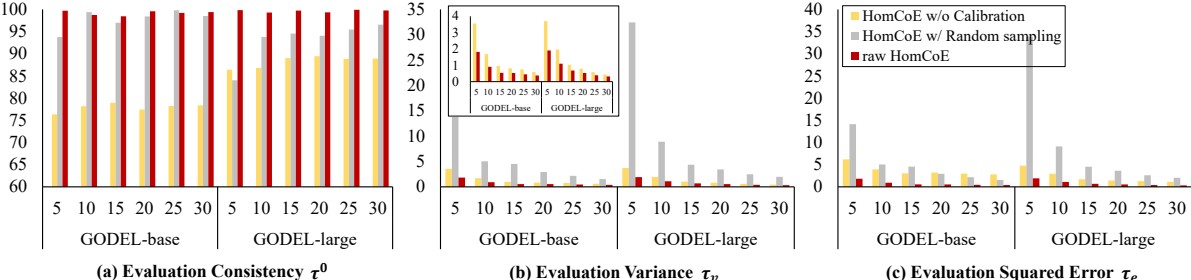

Figure 1: Ablation studies on conversational question answering task. The sampling process enhances the stability of HumCoE, while the alibration procedure focuses more on improving the accuracy.

The reason behind this is that the response of GPT-2 is overestimated by ChatGPT, as evidenced by the higher average relevance score (3.42) compared to human evaluation (3.09). This poses larger weights to selected data during the evaluation calibration, ultimately resulting in less precise evaluation results. However, our method still achieves better results than HMCEval in most cases. On average, it enjoys 7.64% performance gains on Rouge-L, 4.30% on Relevance score, and 10.33% on Naturalness score.

**Summarization on three tasks**. From the above experiments, we have two observations: 1) We experimentally demonstrate that HumCoE produces evaluation results that are highly close to those obtained through human evaluation while also significantly reducing the amount of human labor required. With less than 1% of human labor, HumCoE achieves a consistency rate between 95% and 99% when compared to human evaluation. This reveals that the proposed method in this paper could effectively reduce human labor when evaluating a CIR system. 2) Our approach performs better than other active testing baselines in terms of the average consistency rate for all tasks. Specifically, with less than 1% human labor, our approach achieves an average consistency rate of 98.32%, which is 22.01% higher than the Active Top-k method, 6.51% higher than the HMCEval method, and 6.73% higher than MCM. As mentioned earlier, our method experiences a slight drop in performance when assessing the Relevance score for the GPT-2 model because ChatGPT overestimates the response of GPT-2. However, our approach still showcases its superiority over other baselines across numerous tasks and evaluation metrics.

## 5.2 Ablation studies

This section aims to perform ablation studies on HumCoE and investigate its properties in detail.

We examine the Evaluation Stability (i.e., $\tau_v$ and $\tau_e$) and Evaluation Consistency (i.e., $\tau^\circ$) with or without surrogate machine based data sampling and evaluation calibration. Specifically, we remove surrogate machine based data sampling and replace it with random sampling, which we refer to as *HumCoE w/ Random sample*. *HumCoE w/o Calibration* means no calibration. We experimentally show that our surrogate machine-based sampling enhances the stability of HumCoE, while the calibration procedure focuses more on improving the accuracy.

Due to limited space, we present the results for conversational question answering in Fig.1 and provide the rest in Appendix D. According to $\tau_v$ and $\tau_e$ in Fig.1(b) and Fig.1(c), randomly selecting partial data for human annotations and calibration involves higher variance and instability (i.e., large $\tau_v$ and $\tau_e$). Also, the calibration method enhances stability to some degree. These observations are in line with previous studies stating that the importance sampling is a classic variance reduction technique (Elvira and Martino, 2021). Regarding the evaluation consistency in Fig.1(a), the use of calibration or surrogate machine based data sampling techniques alone tends to reduce the consistency of evaluation results. But even with random sampling, calibration techniques still maintain good results (i.e., *HumCoE w/ Random sample*). This indicates that the calibration technique is more helpful in improving evaluation accuracy.

## 6 Conclusion

Evaluating any highly interactive process is a challenging task when taking human labor into account. To touch upon this challenge, we take the first step to utilize active testing to evaluate the CIR system. We tentatively give a successful implementation to explore the potential of human-machine collaborative evaluation in the CIR scenario. Empirical experiments are conducted to testify to the effec-

tiveness of our method. In a consistent manner, our method produces results that are extremely comparable to those of a human-based evaluation yet require very little human labor compared to other baselines. Our future work may design a more sophisticated active testing method and see the possible outcomes of CIR evaluation.

## 7 Limitations

As demonstrated in Section 5.1.3, some evaluation metrics in CIR cannot be directly computed using a predefined mathematical formula, such as relevance and naturalness scores. This may restrict the application of HumCoE in evaluating CIR. In our experiments, we use ChatGPT to automatically simulate the human scoring process for these two metrics. Although our approach significantly addresses the inconsistency issue between ChatGPT and human evaluations with minimal human effort and provides evaluation results that are highly comparable to human evaluations, we observed a slight reduction in the consistency of our method's evaluation results due to ChatGPT's overestimation when automatically computing the relevant scores of GPT-2. Therefore, when there is no predefined mathematical formula to directly compute an evaluation metric, determining how to further enhance the evaluation accuracy of HumCoE with minimal human labor costs remains a challenge.

## Acknowledgement

This work was supported in part by the National Natural Science Foundation of China (No. 62272330); in part by the Fundamental Research Funds for the Central Universities (No. YJ202219).

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

## A Implementation Details

### A.1 HumCoE Implementation

We use Eq.2 to compute the hardness value $q(d_i)$ for each sample $d_i$. These hardness values are then normalized to a range of 0 to 1 by dividing each element by its sum, i.e., $q(d_i) = q(d_i)/\sum_{d_j} q(d_j)$. To avoid sampling data with excessively small $q(d_i)$, we add a clipping operation by setting $q(d_i) = min(q(d_i), \alpha)$. Here, $\alpha$ is a hyper-parameter, and in practice, we set $\alpha = 0.2/|D_1|$, where $1/|D_1|$ represents the uniform sample probability. We observed that removing excessively small sample weights can enhance evaluation stability. To implement the data sampling step, we use the *WeightedRandomSampler* class in Pytorch. Later, we limit the size of $D_s$ in $\{5, 10, 15, 20, 25, 30\}$ to verify the effectiveness of different evaluation methods with minimal labor costs. Considering the randomness of data sampling, we ran the evaluation 100 times and reported the mean estimation results.

### A.2 Baseline Implementation

There is no previous work on active testing methods for CIR. We re-implement existing baselines as follows. Since our method prefers hard samples, we compare methods that also select hard samples for human annotations, including HMCEval (Zhang et al., 2021a), MCM (Nguyen et al., 2018), and Active Top-K (Rahman et al., 2020). Given that the previous methods are not designed for CIR evaluation, we re-implement them in the CIR setting, where the same entropy-based method is used.

- Most-Confident Mistake (MCM) (Nguyen et al., 2018) selects hard samples whose CIR prediction confidences are very high, and whose CIR prediction results are inconsistent with those of the surrogate machine. For example, if a CIR believes that the current sample is predicted as being positive with high probability, while the surrogate machine's pseudo-label is negative. The label from the surrogate machine is likely at fault, requiring human annotation for correction.

- Active Top-K (Rahman et al., 2020) draws the top-k hard samples from the CIR's ranked uncertainty scores. It has two variants, i.e., Human-Only Active Top-K and Hybrid Active Top-K. The former uses only human-annotated data, while the latter incorporates both human-annotated data and surrogate machine-annotated data for CIR evaluation. In this case, HMCEval (Zhang et al., 2021a) is a Hybrid Active Top-K-based method.

- HMCEval (Zhang et al., 2021a). We have re-implemented HMCEval, originally designed to evaluate malevolence, to fit in the CIR scenario. Specifically, since we treat human labor for each interaction sample equally and machine evaluation is labor-free[5], the solution of HMCEval suggests to assign samples with high prediction certainty to human and machine is responsible for evaluating the rest. To measure uncertainty, we initially used the maximum class probability (MCP), as suggested in the original HMCEval paper. However, MCP is intended for classification-based evaluation metrics (because identifying the malevolence in the dialog is a classification task itself), which does not apply to our first and third tasks which are generation-based. To this end, we resort to the entropy-based method as an alternative to uncertainty modeling. In particular, inspired by (Xiao and Wang, 2019; Liu and Hou, 2023), we calculate the average entropy of generating each word (or tokens) as the uncertainty of the given model generating the whole response. Because the output dimension of the predictive word distribution is very large, it would be imprecise and not very useful to calculate the distribution entropy directly on this high-dimensional distribution. Therefore, we choose the top-5 values from the predictive distribution and convert them into a 5-dimensional distribution using Soft-Max (We also use the same trick for MCM and Active Top-K). We then use this to calculate the word entropy.

### A.3 Task Setup & Task Model Implementation

In HumCoE, we train all surrogate machines using the same training data as the CIR model $f$. We

---

[5]Treating human labor equally on each data point is widely used and necessary in current research on Active testing (Nguyen et al., 2018; Rahman et al., 2020), CIR (Alianne-jadi and Trippas, 2022; Lei et al., 2020a), and human-machine interaction (Klie et al., 2020; Desmond et al., 2021).

Table 4: Automatic calculation on relevance and naturalness score using ChatGPT scoring

| Prompt Template |
| --- |
| 1. [QUESTION] I am think about ${facet}. I say that ${query}. [ANSWER] ${answer}. [SCORE] {"Relevence": ?, "Naturalness": _} |
| 2. [QUESTION] I am think about ${facet}. I say that ${query}. [ANSWER] ${answer}. [SCORE] {"Relevence": ?, "Naturalness": _} |
| ### On a 5-scale, 5 is the best, rate the [ANSWER] in above sentence based on the following criteria: |
| |
| Relevence: Is the [ANSWER] relevant to the [QUESTION]. |
| |
| Naturalness: Is the [ANSWER] natural. |
| |
| Here are some examples: |
| 1) [QUESTION] I am think about shirts. I say that Find Brooks Brothers clearance. [ANSWER] are you looking for a specific Brooks Brothers. [SCORE] {"Relevence": 2, "Naturalness": 1} |
| |
| 2) [QUESTION] I am think about list homes sale. I say that tell me about cass county missouri. [ANSWER] are you looking for a sale for a home in dellas. [SCORE] {"Relevence": 4, "Naturalness": 4} |
| |
| Present the scores in JSON format as follows: |
| {"Relevence":<float>, "Naturalness":<float>} |
| Please provide scores and explain reasons.### |

conducted all our experiments on an NVIDIA RTX A6000 GPU with 48G graphical memory.

### A.3.1 Conversational Question Answering

Conversational question-answering systems need to handle the more complex linguistic characteristics of conversations, anaphoras or ellipses may occur in question sentences (Vakulenko et al., 2021). In user-CIR conversations, the CIR system must respond to every question posed by the user. As for the task model in HumCoE, we use the recently proposed and open-sourced method GODEL (Peng et al., 2022), which can be downloaded from Huggingface[6], and we use the given instruction to finish a conversational question answering task. In HumCoE, we use GODEL-base as the task model $f$ and GODEL-large as the surrogate machine $g$, and vice versa. Following previous works (Kim et al., 2022; Peng et al., 2022), we use the CoQA dataset (Reddy et al., 2019) for conversational question-answering simulation. Based on the test data, we compared the answers generated by the GODEL-base and GODEL-large on the test data with the manually annotated answers, respectively. We use Rouge-L to assess the word overlap between the prediction and the ground truth answer. In this case, $q(d_i) \propto 1 - \text{Rouge-L}(d_i, \hat{y}_i)$. Note that human evaluation of the two models yields results of 68.56 and 75.09.

### A.3.2 Clarifying Question (Retrieved)

The CIR system needs to have a clear understanding of the underlying user needs. Since user's queries are often under-specified and vague, with a mixed-initiative conversational search, the CIR can take the lead and ask the user clarifying questions to clear up the ambiguity in the user's query

---

[6]GODEL-large: https://huggingface.co/microsoft/GODEL-v1_1-large-seq2seq, GODEL-base: https://huggingface.co/microsoft/GODEL-v1_1-base-seq2seq

(Sekulić et al., 2022; Keyvan and Huang, 2022). Formally, the user expresses their demand in the initial query form, which is subsequently provided to the CIR system. The user needs to generate an answer to the system's clarifying question. The CIR system aims to elucidate the information need through a series of clarifying questions and return the correct answer. Following (Aliannejadi et al., 2020), we assume that the clarifying questions are retrieved from a pre-defined question pool. In this task, we evaluate the task success rate (i.e., if it returns the correct answer at the end), and ignore the evaluation of question quality as the question is pre-defined. Regarding the task model in this task, we follow the previous study (Aliannejadi et al., 2021), and utilize BERT-base and ERNIE-base as $f$ and $g$ interchangeably in our experiments. These two models are fine-tuned on ClariQ (Aliannejadi et al., 2021, 2020) training data and then evaluated on its test data. Specifically, we fine-tune each model on one epoch with $5e^{-7}$ learning rate, 32 batch size, and Adam optimizer. Considering that it is a classification problem, we utilize the accuracy metric to measure the task success rate. In this case, $q(d_i) \propto 1 - \text{Acc}(d_i, \hat{y}_i)$. Note that the results of human evaluation on the BERT and ERNIE are 88.03% and 91.79%, respectively.

### A.3.3 Clarifying Question (Generated)

In this task, the CIR system that *generates* clarifying questions is evaluated. Here, we shift the focus to evaluating the quality of the generated questions. Following earlier studies (Aliannejadi et al., 2021, 2020), we also consider ClariQ in our experiments. To evaluate the quality of the generated questions, we consider two types of evaluation metrics, i.e., automatic and user-based metrics. Rouge-L is used as the automatic evaluation metric, while the user-based evaluation is conducted by the human providing ground truth (cf. Appendix C) and the ChatGPT

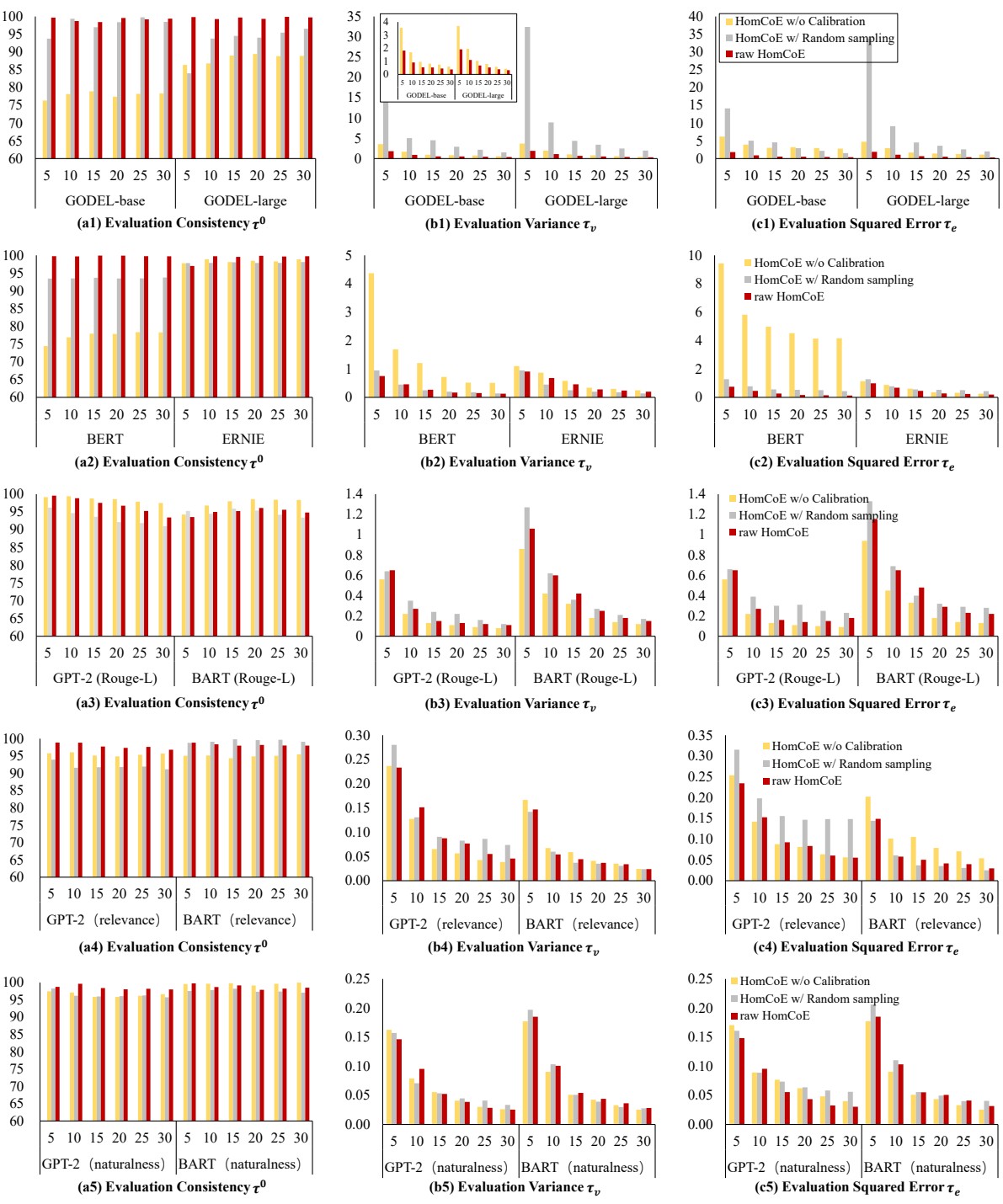

Figure 2: Ablation studies on all tasks and metrics.

completing the automatic evaluation (cf. Appendix B). In this case, $q(d_i) \propto 1 - \text{Rouge-L}(d_i, \hat{y}_i)$ and $q(d_i) \propto 1 - \text{ChatGPT}(d_i, \hat{y}_i)$. Here, we report the results of human evaluation on the BART and GPT-2: the Rouge-L scores on the BART and GPT-2 are 37.57 and 48.46, respectively. Meanwhile, the relevance score and naturalness score for GPT-2 are 3.09 and 3.51, respectively, while the scores

for BART are 3.82 and 3.85, respectively. As for the task model, we follow (Sekulić et al., 2021) and alternate between using BART-base and GPT2-base, which are also fine-tuned on ClariQ training. Specifically, we fine-tune each model for 5 epochs using an Adam optimizer, $5e^{-5}$ learning rate, and a batch size of 32.

## B    Automatic calculation on relevance and naturalness score using ChatGPT

Tasks (such as conversational question-answering) utilize automatic metrics (such as Rouge-L) for evaluation. In terms of evaluating model-generated-based clarifying questions, relevance and naturalness cannot be automatically calculated. To this end, we resort to ChatGPT to calculate relevance and naturalness. As ChatGPT requires a fee and has been banned in some countries, we use Monica[7] for annotations, which is a Chrome extension powered by ChatGPT API. As free users of Monica, we have a daily usage limit. Thus, we require Monica to annotate multiple data at a time (precisely, 20 at a time). Specifically, to enhance the precision of ChatGPT scoring, we include two score definitions and scoring cases as input. The prompt template for ChatGPT scoring can be found in Table 4.

## C    User-evaluation on relevance and naturalness score (human annotations)

For the clarifying question task with model-generated questions, we follow previous work (Aliannejadi et al., 2020) and employ 3 annotators to assess the relevance and naturalness of each system response (i.e., GPT-2 and BART) on the test data. The relevance score (0–5) measures if the clarifying question is relevant to the user's information needs, while the naturalness score (0–5) measures if the clarifying question is natural in the context of the conversation.

In particular, we provide each annotator with the meanings and evaluation protocols of two scores. Three annotators independently annotate and evaluate the system responses on the test data, discuss any disagreements, and revise the annotation scheme. Before annotating the test data, the three annotators complete the annotation exercises on 100 examples randomly selected from the training data. They continue this process until they achieve inter-annotator reliability of Krippendorff's alpha greater than 0.7 on these samples. Afterward, each annotator scores all the system responses. The final human evaluation score for each data is the mean of the scores given by the three annotators. This approach improves the robustness of the results.

---
[7] https://monica.im/

Table 5: Evaluation Consistency on TopiOCQA.

| CIR | # | HumCoE (ours) | Active Top-K | HMCEval | MCM |
|---|---|---|---|---|---|
| DRP | 5 | $82.87_{(3.51)}$ | $17.05_{(16.98)}$ | $76.19_{(4.87)}$ | $20.90_{(16.19)}$ |
| | 10 | $77.73_{(4.56)}$ | $19.43_{(16.49)}$ | $76.19_{(4.87)}$ | $23.80_{(15.59)}$ |
| | 15 | $82.68_{(3.54)}$ | $32.58_{(13.80)}$ | $76.19_{(4.87)}$ | $21.30_{(16.11)}$ |
| | 20 | $80.94_{(3.90)}$ | $24.43_{(15.47)}$ | $76.00_{(4.91)}$ | $23.70_{(15.62)}$ |
| | 25 | $79.81_{(4.13)}$ | $39.09_{(12.47)}$ | $76.00_{(4.91)}$ | $22.10_{(15.94)}$ |
| | 30 | $79.55_{(4.19)}$ | $32.58_{(13.80)}$ | $76.00_{(4.91)}$ | $23.60_{(15.64)}$ |
| FID | 5 | $90.18_{(4.25)}$ | $30.33_{(30.19)}$ | $72.70_{(11.83)}$ | $15.40_{(36.66)}$ |
| | 10 | $91.98_{(3.47)}$ | $45.26_{(23.72)}$ | $72.79_{(11.79)}$ | $7.70_{(40.00)}$ |
| | 15 | $90.89_{(3.95)}$ | $50.28_{(21.55)}$ | $72.78_{(11.80)}$ | $3.80_{(41.69)}$ |
| | 20 | $89.41_{(4.59)}$ | $65.35_{(15.01)}$ | $72.82_{(11.78)}$ | $10.60_{(38.74)}$ |
| | 25 | $90.62_{(4.07)}$ | $72.29_{(12.01)}$ | $72.85_{(11.77)}$ | $5.40_{(41.00)}$ |
| | 30 | $90.61_{(4.07)}$ | $64.11_{(15.55)}$ | $72.83_{(11.77)}$ | $9.20_{(39.35)}$ |

## D    Appendix on ablation studies

We report the results of ablation studies in Fig. 2, which are consistent with the previous observations in Section 5.2. However, we notice some special cases in the results. For example, in Fig. 2(b2) and Fig. 2(c2), without calibration, the evaluation stability is worse than the case without surrogate machine based data sampling. This motivates more future work to study better and more stable active testing algorithms.

However, from the overall experimental results, we experimentally show that our surrogate machine-based data sampling enhances the stability of HumCoE, while the calibration procedure focuses more on improving our accuracy. According to the results of evaluation consistency $\tau^\circ$, combining evaluation calibration and surrogate machine based data sampling methods can achieve better results in terms of the average consistency rate for all tasks. Specifically, our approach achieves an average consistency rate of 98.32%, which is 2.75% higher than the *HumCoE w/ Random sample* and 5.88% higher than the *HumCoE w/o Calibration*. It also implies that calibration brings 3.13% performance gains over *HumCoE w/o Calibration*. Regarding the evaluation stability, our HumCoE also enjoys better results in terms of the average $\tau_v$ and average $\tau_e$ for all tasks. In particular, our approach achieves an average $\tau_v$ of 0.34, which translates into 1.24 higher than the *HumCoE w/ Random sample* and 0.24 higher than the *HumCoE w/o Calibration*. We also achieve an average $\tau_e$ of 0.35, which is 1.36 higher than the *HumCoE w/ Random sample* and 0.95 higher than the *HumCoE w/o Calibration*. These indicate that the surrogate machine based data sampling method brings higher evaluation stability than the random one.

# E Appendix on Related Work

**Human-machine Evaluation.** Human-machine evaluation has been under-explored in the NLP community, where the traditional focus has been on building a better model (Wang et al., 2021). They mainly build upon the human-in-the-loop scenario, where the values of automatic evaluation metrics are provided for humans as evaluation suggestions (Khashabi et al., 2021; Sedoc et al., 2019). Such methods only reduce human cognitive labor yet still require humans to go through the whole dataset. We argue that they are machine-assisted evaluation methods rather than human-machine collaboration methods for evaluation. Thus, current human-machine evaluation is out of our research scope.

**Learning to defer**. One way to achieve human-machine evaluation is through learning to defer (Madras et al., 2018; Mozannar and Sontag, 2020), also known as rejection learning (Chow, 1970). This approach involves assigning tasks to either machines or humans to make decisions. When applied to the evaluation task, the main goal of learning to defer is to determine which data should be given to humans and which should be given to machines to ensure accurate evaluation results for the entire dataset. Similar to HMCEval (Zhang et al., 2021b), the most common method is to assign data with high machine prediction uncertainty to humans and easier-to-predict data to machines (Ni et al., 2019; Grandvalet et al., 2008) However, our method differs from learning to defer in that we focus on selecting a subset of data for human evaluation that can accurately estimate the evaluation results for the entire dataset, without requiring machines to evaluate the remaining data.

# F Appendix on More Experiments

To provide a more comprehensive understanding, we conduct additional experiments on TopiOCQA (Adlakha et al., 2022), where the CIR model is DPR (Karpukhin et al., 2020) and the surrogate model is FiD (Izacard and Grave, 2021) (which are used interchangeably). Here, both models are fine-tuned on the TopiOCQA dataset. The results, shown in Table 5, demonstrate that our HumCoE approach continues to achieve promising results and outperforms the baselines.