# OpenReview forum: "Reduce Human Labor On Evaluating Conversational Information Retrieval System: A Human-Machine Collaboration Approach"
_EMNLP/2023/Conference — EMNLP 2023 Main_

### Official Review · Reviewer_ReMS · 2023-08-02

**Soundness:** 4

**Excitement:**

4: Strong: This paper deepens the understanding of some phenomenon or lowers the barriers to an existing research direction.

**Paper Topic And Main Contributions:**

This paper proposes a human-machine collaborative approach called HumCoE to reduce the amount of human labor needed for evaluating conversational information retrieval (CIR) systems. They propose to utilize active testing for CIR evaluation by strategically selecting representative data for human annotation. This is the first work to apply active testing for CIR evaluation.
The proposed HumCoE method selects data based on the probability of inconsistency between the CIR model prediction and pseudo labels from a surrogate model.
Empirically results demonstrate that HumCoE produces evaluation results comparable to full human evaluation with minimal human labor (<1%) across different CIR tasks like conversational QA and clarifying questions.

**Questions For The Authors:**

For the clarifying question generation task, ChatGPT is used to automatically predict relevance/naturalness scores. But the agreement of ChatGPT with human scores was low based on Krippendorff's alpha. Could the authors comment more on the reliability of using ChatGPT for automatic scoring?

Have you experimented with different surrogate model architectures? Does the choice of surrogate model significantly affect HumCoE performance?

What are some scenarios where HumCoE does not approximate full human evaluation well? Is there a theoretical analysis of the lower bounds on human labeling needed?

**Reasons To Accept:**

1. Addresses the important challenge of designing labor-efficient evaluation methods for conversational systems. Reducing human annotation effort would have significant practical impact.

2. Proposes a novel approach by combining ideas from active learning and human-AI collaboration. Adaptation of active testing for CIR is innovative.

3. Comprehensive empirical analysis on multiple datasets and CIR tasks demonstrates the effectiveness over baselines. The method consistently approximates full human evaluation with minimal labor.

The problem is timely and the solution is novel. The approach could inspire more work on human-machine collaboration for efficient system evaluation.

**Reasons To Reject:**

The weaknesses are minor compared to the contributions. One limitation is the need for a differentiable evaluation metric to estimate sample hardness (relevance scores addressed via ChatGPT).

The authors could do more error analysis and discuss scenarios where the approach may not work as well. But overall the claims seem properly supported.



**Reproducibility:**

3: Could reproduce the results with some difficulty. The settings of parameters are underspecified or subjectively determined; the training/evaluation data are not widely available.

**Reviewer Confidence:**

1: Not my area, or paper was hard for me to understand. My evaluation is just an educated guess.

**Typos Grammar Style And Presentation Improvements:**

Section 2: "CIR evaluation methods is still an unaddressed problem" -> "CIR evaluation methods are still an unaddressed problem"

Section 5.1.3: "we also consider ClariQ in our experiments" -> "we also consider ClariQ dataset in our experiments"

Figure 1: Subcaptions are not readable. Please make them larger.

Some minor grammar/style issues exist but overall the paper is well written.

---

> ### Author Rebuttal · Authors · 2023-08-28
>
> We thank you for your detailed feedback and suggestions. In the following, we clarify your concerns in order.
>
> # Regarding "Reasons To Reject"
> We appreciate the reviewer's positive feedback and valuable suggestions. As mentioned, ChatGPT offers a promising alternative to address the challenge of requiring differentiable evaluation metrics. We will explore other non-differentiable evaluation metrics using ChatGPT to assess the effectiveness more broadly. Additionally, we plan to conduct further error analysis and include it in the journal extension once our work is accepted.
>
> # Could the authors comment more on the reliability of using ChatGPT for automatic scoring? (Q1)
> As shown in our study (Line 537-565), the consistency between ChatGPT-generated scores and human scores for relevance and naturalness is not high, which aligns with recent findings that point-wise scoring by ChatGPT is unreliable[1]. This suggests that ChatGPT alone may not be a reliable tool for automatic CIR evaluation, at least for relevance and naturalness scoring. One possible way to improve the reliability of ChatGPT's scores is to use pair-wise and list-wise ranking, treating the ranking index as the score for the dataset. However, in our case, we require ChatGPT to provide specific scores (0-5) to align with the manual scores as much as possible. This implies that the point-wise approach is currently the best solution for us to achieve automated evaluation. Fortunately, our method does not rely solely on ChatGPT scores to calculate evaluation results; instead, we use these scores in the data sampling step. The experimental results demonstrate that our method achieves high consistency between human evaluation even with the imperfect scores generated by ChatGPT.
> [1] Uncovering ChatGPT's Capabilities in Recommender Systems, 2023
>
> # Have you experimented with different surrogate model architectures? Does the choice of surrogate model significantly affect HumCoE performance? (Q2)
>
> In our experiments, We explored surrogate models that differ from the CIR model, including BART and GPT2, BERT, and ERNIE. We are excited to present further discussions and results on this topic.
>
> **More Discussions**. We believe that using surrogate models with different model architectures fundamentally promotes diversity [1]. As mentioned in Lines 246-250, it is crucial to encourage diversity between the CIR model f and the surrogate model g. To illustrate this, consider an extreme example: Our method relies on g to provide pseudo-labels, which allows us to estimate the probability of f making mistakes on each data point and select data with high probabilities of being errors. If f and g are the same, we would not be able to accurately select data since the predictions of g and f would be identical, and the probability of error for each data point would be zero. In such cases, our method would randomly select data without calibration. Therefore, we recommend using different surrogate models when evaluating CIR systems.
>
> **More results**. We conducted additional experiments using different surrogate models (i.e., **g**) to evaluate the same CIR model. We focused on the clarifying question task and used ERNIE, ELECTRA, and DeBERTa as surrogate models. The results demonstrate that: 1) our HumCoE approach outperforms baselines, 2) HumCoE consistently outperforms different surrogate models, and 3) surrogate models with larger model parameters do not necessarily lead to higher performance in terms of evaluation consistency.
>
> | **CIR**         | **#** | **Active Top-K** | **HMCEval** | **MCM** | **Ours (g=ERNIE-base)** | **Ours (g=ERNIE-large)** | **Ours (g=DeBERTa-base)** | **Ours (g=DeBERTa-large)** | **Ours (g=ELECTRA-base)** | **Ours (g=ELECTRA-large)** |
> |-----------------|-------|------------------|-------------|---------|-------------------------|--------------------------|---------------------------|----------------------------|---------------------------|----------------------------|
> | BERT            |     5 | 90.88            | 97.52       | 97.52   | **99.76**               |                **98.44** |                 **98.56** |                  **98.60** |                 **98.29** | **98.43**                  |
> |                 | 10    | 56.80            | 97.52       | 97.52   | **99.69**               |                **99.20** |                 **99.13** |                  **99.46** |                 **98.81** | **99.21**                  |
> |                 | 15    | 53.02            | 97.50       | 97.50   | **99.93**               |                **99.32** |                 **99.38** |                  **99.64** |                 **99.12** | **99.44**                  |
> |                 | 20    | 51.12            | 97.40       | 97.40   | **99.91**               |                **99.56** |                 **99.56** |                  **99.82** |                 **99.11** | **99.47**                  |
> |                 | 25    | 59.07            | 97.32       | 97.32   | **99.75**               |                **99.79** |                 **99.80** |                  **99.77** |                 **99.45** | **99.82**                  |
> |                 | 30    | 60.58            | 97.32       | 97.30   | **99.73**               |                **99.74** |                 **99.63** |                  **99.34** |                 **99.97** | **99.61**                  |
> | avg.performance | -     |            61.91 |       97.43 |   97.43 |               **99.80** |                **99.34** |                 **99.34** |                  **99.44** |                 **99.12** | **99.33**                  |
>
> [1] A Unified Theory of Diversity in Ensemble Learning
>
>
> # What are some scenarios where HumCoE does not approximate full human evaluation well? Is there a theoretical analysis of the lower bounds on human labeling needed? (Q3)
>
> Theoretically, we will try to determine the theoretical bound for our method in the future. We believe diversity analysis may be a good place to start. But, this still requires more effort.
>
> Empirically, surrogate models with very low diversity (discussed above) or poor performance can undermine the advantages of our method. For instance, if the surrogate model provides pseudo-labels randomly, the calibration step may not function as intended. Nevertheless, it is not difficult to find a surrogate model with good performance and high diversity for evaluating CIR systems, given the abundance of publicly available LLMs. Therefore, we intentionally used an open-source language model in our experimental study, which is more aligned with real-world scenarios and eliminates the need to search for a suitable surrogate model.

---

### Official Review · Reviewer_sHgf · 2023-08-04

**Soundness:** 3

**Excitement:**

3: Ambivalent: It has merits (e.g., it reports state-of-the-art results, the idea is nice), but there are key weaknesses (e.g., it describes incremental work), and it can significantly benefit from another round of revision. However, I won't object to accepting it if my co-reviewers champion it.

**Paper Topic And Main Contributions:**

The paper proposes a method for active testing of model generations named HomCOE. The method actively selects a few model generations to be annotated by humans , to limit the significant cost of human labor for manual evaluation.

The methodology selects representative data for human annotation by factoring in the hardness of examples. Hardness of examples are measured as the probability of the model making mistakes against pseudo labels. To moderate the effect of only selecting hard examples for evaluation they balance with sample weights.

**Reasons To Accept:**

The paper studies an important and timely problem of evaluating model generations with a limited budget for human annotations.

Evaluations are conducted on multiple tasks such as Conversational Question Answering and generation of Clarifying Questions. Experimental results demonstrate they improve over other baselines for selecting hard samples.

**Reasons To Reject:**

The proposed method has limited novelty and appears incremental compared to HMCEval. Additionally, treating each sample equally for human labor cost might be a strong assumption.

Would also recommend to validate the performance of the  approach on few other conversational datasets at least.

**Reproducibility:**

3: Could reproduce the results with some difficulty. The settings of parameters are underspecified or subjectively determined; the training/evaluation data are not widely available.

**Reviewer Confidence:**

2: Willing to defend my evaluation, but it is fairly likely that I missed some details, didn't understand some central points, or can't be sure about the novelty of the work.

---

> ### Author Rebuttal · Authors · 2023-08-28
>
> Thank you for your feedback. However, we cannot agree with the main points of your criticism. Please see our detailed response to your "Reasons To Reject". Hope we can clarify your misunderstanding.  If any part of our description is unclear, please don't hesitate to ask us for clarification.
>
> # 1.Limited novelty and appears incremental compared to HMCEval (R1)
> As noted by ourselves and other reviewers, accurately evaluating a CIR system at a low cost is an urgent problem that requires attention. In response to this problem, we made a successful attempt and implementation based on our proposed active testing method. In addition to the innovation inherent in our method, we hope that the reviewers will pay more attention to the significance of our paper in solving the problem of the CIR evaluation.
>
> Having experimentally shown our superiority over HMCEval, we would also like to highlight that our method is largely different from HMCEval in terms of the methodology, optimization strategy, and targeted evaluation task, although we are both human-machine collaborative methods for evaluation. We will provide more explanations to delve deeper into their differences.
>
> **Methodology**.  1) **HMCEval** is based on the human-machine collaboration paradigm of **learning to defer**[1-2] (or called rejection learning [3]), where tasks are jointly fulfilled by the machine or deferred to humans in making decisions. HMCEval takes the first step to involve learning to defer to model evaluation task, it requires humans to evaluate a subset of data and machines to evaluate the remaining data. The primary research question in HMCEval, as well as the learning-to-defer studies, is to decide which data should be assigned to humans and which to machines to achieve accurate evaluation results for the whole dataset. The most common approach [4,5] is to assign data with high machine prediction uncertainty to humans, while assigning the rest, which are easier to predict, to machines. This is also the approach adopted by HMCEval. 2) **Our method** follows the human-machine collaboration paradigm of **active testing** [6], where humans evaluate a subset of data. Our research question, as well as the active testing studies, is how to select a subset of data for human evaluation that accurately estimates the evaluation results for the entire dataset. Unlike HMCEval, our method does not require machines to evaluate the remaining data. Instead, we use machines to assist with subset selection. Additionally, Unlike other active testing methods[6,7], our approach incorporates machine assistance to calibrate the evaluation results so that it improves evaluation performance.
>
> **Optimization strategy**. 1) HMCEval formulates a constrained optimization problem (cf. Lines 306-310); 2) We directly sample representative data based on the estimated probabilities, following the active testing studies (cf. Section 4).
>
> **Targeted Evaluation Tasks**. HMCEval is designed for malevolence identification evaluation, while our method aims at evaluating the conversional information retrieval system (cf. Lines 184-188, 306-310, 968-971).
>
> # 2.Treating each sample equally for human labor cost might be a strong assumption (R1)
> We want to emphasize that the method of treating human labor equally on each data point is widely used and necessary in current research on Active testing [6,7,8], CIR [9,10,11], and human-machine interaction [12-15]. While it would be preferable to have a more detailed estimation of human labor, it is challenging as human labor is not only influenced by the difficulty of the data but also by the complexity of human behavior, as recognized in many IR/CIR studies [16]. Therefore, current studies typically treat human labor equally on each data point as a compromise. We also follow this common practice in our work.
>
> # 3.Recommend to validate the performance of the approach on few other conversational datasets (R2)
> Based on a recent CIR survey [11], conversational QA and clarifying questions are common use cases of CIR, both of which we evaluated in our paper. We validated the effectiveness of our method on different CIR tasks, models, and evaluation metrics, as acknowledged by Reviewer ReMS. However, given the vast scope of research in the CIR field, it would be challenging for us and other researchers to conduct experiments on all tasks. To provide a better understanding, we conducted additional experiments based on the suggestions of Reviewer Hngp. We present the results on TopiOCQA [5], where the CIR model is DPR [6] and the surrogate model is FiD [7] (which are used interchangeably). The results demonstrate that our HumCoE approach continues to achieve promising results and outperforms the baselines.
>
> | **CIR** | **#** | **HumCoE (ours)** | **Active Top-K** | **HMCEval** | **MCM** |
> |---------|-------|-------------------|------------------|-------------|---------|
> |   DPR   |     5 |        **82.87%** |           17.05% |    76.19%   |  20.90% |
> |         |    10 |        **77.73%** |           19.43% |  76.19%     |  23.80% |
> |         |    15 |        **82.68%** |           32.58% |      76.19% |  21.30% |
> |         |    20 |        **80.94%** |           24.43% |      76.00% |  23.70% |
> |         |    25 |        **79.81%** |           39.09% |      76.00% |  22.10% |
> |         |    30 |        **79.55%** |           32.58% |      76.00% |  23.60% |
> |   FID   |     5 |        **90.18%** |           30.33% |      72.70% |  15.40% |
> |         |    10 |        **91.98%** |           45.26% |      72.79% |   7.70% |
> |         |    15 |        **90.89%** |           50.28% |      72.78% |   3.80% |
> |         |    20 |        **89.41%** |           65.35% |      72.82% |  10.60% |
> |         |    25 |        **90.62%** |           72.29% |      72.85% |   5.40% |
> |         |    30 |        **90.61%** |           64.11% |      72.83% |   9.20% |
>
> **Reference**
> - [1] Improving Fairness and Accuracy by Learning to Defer, 2018
> - [2] consistent estimators for learning to defer to an expert, 2020
> - [3] on optimum recognition error and reject tradeoff, 1970
> - [4] on the calibration of multiclass classification with rejection, 2019
> - [5] support vector machine with a reject option, 2008
> - [6] Active testing: An efficient and robust framework for estimating accuracy, 2018
> - [7] Efficient test collection construction via active learning, 2020
> - [8] Active Surrogate Estimators: An Active Learning Approach to Label-Efficient Model Evaluation, 2022
> - [9] Conversational Recommendation: Formulation, Methods, and Evaluation, 2020
> - [10] Conversational Information Seeking: Theory and Evaluation: CHIIR 2022 Half Day Tutorial, 2022
> - [11] Neural Approaches to Conversational Information Retrieval, 2022
> - [12] Interactive information extraction with constrained conditional random fields, 2004
> - [13] MATILDA - Multi-AnnoTator multi-language InteractiveLight-weight Dialogue Annotator, 2021, EACL
> - [14] From Zero to Hero: Human-In-The-Loop Entity Linking in Low Resource Domains, 2020, ACL
> - [15] Semi-Automated Data Labeling, 2021, NeurIPS
> - [16] User behavior modeling for Web search evaluation, 2020

---

### Official Review · Reviewer_sq7x · 2023-08-12

**Typos Grammar Style And Presentation Improvements:** Line 349 – “very few cost” → very low…
**Soundness:** 4

**Excitement:**

3: Ambivalent: It has merits (e.g., it reports state-of-the-art results, the idea is nice), but there are key weaknesses (e.g., it describes incremental work), and it can significantly benefit from another round of revision. However, I won't object to accepting it if my co-reviewers champion it.

**Paper Topic And Main Contributions:**

This paper provides a new evaluation method for the CIR task that makes use of active testing techniques to reduce the amount of needed human annotation.  They first select a small set of data to be annotated (found using scoring with a surrogate model) and then weight the final set with a calibration technique from active learning literature.  They compare against a handful of similar active testing approaches and note how theirs compares favorably with human evaluation on this task using consistency and stability metrics.

--------
Note: I've adjusted my soundness scores during the discussion period.  See discussion with authors regarding main concerns below.

**Reasons To Accept:**

- Detailed experiments on three CIR tasks with promising improvement over baselines
- first approach at leveraging this type of active testing technique for CIR tasks

**Reasons To Reject:**

- The methods section (section 4) was a bit confusing and this may have affected my ability to judge the soundness of the approach.  Some of the variables and equations need to be explained more fully.  In particular, I think there needs to be more explanation given about the usage of the surrogate model.  A few running examples would probably help clarify this.
- I'm also less familiar with the evaluation metrics that are being used here (the stability and consistency ones) and I think that they should be explained a bit more.  Authors could provide references to other works that use these or they could just explain them in more depth.
- How sensitive are the final humcoe scores to the choice of surrogate model g? I would like to see more discussion or experiments showing how using different models for g (with the same model for f) causes scores to vary.

**Reproducibility:**

2: Would be hard pressed to reproduce the results. The contribution depends on data that are simply not available outside the author's institution or consortium; not enough details are provided.

**Reviewer Confidence:**

3: Pretty sure, but there's a chance I missed something. Although I have a good feel for this area in general, I did not carefully check the paper's details, e.g., the math, experimental design, or novelty.

---

> ### Author Rebuttal · Authors · 2023-08-28
>
> We appreciate your feedback and suggestions. In Section 4, we made an effort to provide detailed explanations of our method. However, based on your feedback, we understand that we may not have been able to convey all the details to every reviewer. We hope that our subsequent responses will address your concerns and provide a sufficient understanding of our approach. If any part of our description is unclear, please don't hesitate to ask us for clarification.
>
> # 1.More explanation on methodology, especially on surrogate model (R1)
> Our approach consists of two main steps: data sampling and evaluation calibration. In the data sampling step, we utilize a surrogate model as a user proxy to generate pseudo-labels that aid in selecting representative data for human evaluation.
>
> To illustrate, let's consider a scenario where we have ten conversational QA data. In the human evaluation process, the human first annotates the ground truth answers and assesses the accuracy of the system's predicted answers. However, due to resource constraints, we now assume that we can only afford to evaluate three of them. To ensure the evaluation results accurately reflect the whole dataset, we aim to select three representative data points that are hard for the CIR system. To achieve this, we employ the surrogate model to generate pseudo answers for all ten data points, allowing us to estimate the probabilities of the CIR system making mistakes without requiring ground truth answers from humans. Based on these probabilities, we sample three representative data and then require for human evaluation. As for the evaluation calibration, to mitigate the potential bias introduced by evaluating only a subset of the data, we incorporate a data weight (i.e., Eq. 3) to calibrate the evaluation results. This helps reduce the impact of hard data on the overall evaluation. Our experiments demonstrate that this simple yet effective method yields promising results.
>
>
> # 2.More explanation on metrics (R2)
> Inspired by [1, 2], we assess the performance from two aspects: evaluation consistency and stability (cf. Line 347-366, page 4). Recalling the ten-data example above, evaluation consistency assesses whether the evaluation results, obtained from evaluating three selected data, can accurately represent the evaluation results from the entire dataset. Since our method and baselines involve data sampling, which introduces randomness, we run experiments multiple times and use the averaged evaluation results to measure consistency (i.e., Eq. 5). We further consider measuring the stability of the evaluation results across different runs. We use two metrics, namely variance and squared error of inconsistency of each run, to achieve this. This allows us to identify potential sources of instability.
>
> - [1] Active Testing: An Efﬁcient and Robust Framework for Estimating Accuracy, 2018
> - [2] Active Surrogate Estimators: An Active Learning Approach to Label-Efficient Model Evaluation, 2022
>
> # 3.Sensitivity to the choice of surrogate model (R3)
> Thanks for your suggestions. In our paper, we conducted preliminary experiments on the sensitivity of our method to different surrogate models, as shown in Tables 1, 2, and 3. Specifically, we tested our approach using different CIR models and surrogate models interchangeably. The results indicate that our method is effective and robust across different CIR models and surrogate models.
>
> However, to address your concerns, we conducted additional experiments using different surrogate models (i.e., **g**) to evaluate the same CIR model. Due to time constraints, we focused on the clarifying question task and used ERNIE, ELECTRA, and DeBERTa as surrogate models. The results demonstrate that: 1) our HumCoE approach outperforms baselines, 2) HumCoE consistently outperforms different surrogate models, and 3) surrogate models with larger model parameters do not necessarily lead to higher performance in terms of evaluation consistency.
>
> | **CIR**         | **#** | **Active Top-K** | **HMCEval** | **MCM** | **Ours (g=ERNIE-base)** | **Ours (g=ERNIE-large)** | **Ours (g=DeBERTa-base)** | **Ours (g=DeBERTa-large)** | **Ours (g=ELECTRA-base)** | **Ours (g=ELECTRA-large)** |
> |-----------------|-------|------------------|-------------|---------|-------------------------|--------------------------|---------------------------|----------------------------|---------------------------|----------------------------|
> | BERT            |     5 | 90.88            | 97.52       | 97.52   | **99.76**               |                **98.44** |                 **98.56** |                  **98.60** |                 **98.29** | **98.43**                  |
> |                 | 10    | 56.80            | 97.52       | 97.52   | **99.69**               |                **99.20** |                 **99.13** |                  **99.46** |                 **98.81** | **99.21**                  |
> |                 | 15    | 53.02            | 97.50       | 97.50   | **99.93**               |                **99.32** |                 **99.38** |                  **99.64** |                 **99.12** | **99.44**                  |
> |                 | 20    | 51.12            | 97.40       | 97.40   | **99.91**               |                **99.56** |                 **99.56** |                  **99.82** |                 **99.11** | **99.47**                  |
> |                 | 25    | 59.07            | 97.32       | 97.32   | **99.75**               |                **99.79** |                 **99.80** |                  **99.77** |                 **99.45** | **99.82**                  |
> |                 | 30    | 60.58            | 97.32       | 97.30   | **99.73**               |                **99.74** |                 **99.63** |                  **99.34** |                 **99.97** | **99.61**                  |
> | avg.performance | -     |            61.91 |       97.43 |   97.43 |               **99.80** |                **99.34** |                 **99.34** |                  **99.44** |                 **99.12** | **99.33**                  |

---

### Official Review · Reviewer_Hngp · 2023-08-14

**Soundness:** 4

**Excitement:**

4: Strong: This paper deepens the understanding of some phenomenon or lowers the barriers to an existing research direction.

**Paper Topic And Main Contributions:**

This work mainly addresses the problem of the automatically evaluation for conversational information retrieval, providing a landmark and more labor-effective methods
for future human-machine-based CIR evaluation research.

The main contributions are:

1. Introduce active testing to evaluate CIR system and propose a novel method, called HumCoE, based on selecting representative data for human annotation and carefully calibrating the evaluation results, based on surrogate machine.

2. Verify the effectiveness with empirical studies. The results show that it makes an accurate evaluation at low human labor.

To sum up, this paper tries to address an important and interesting problem, which is valuable for the CIR research community development. The proposed method is promising, and the experiments seems supportive.

**Questions For The Authors:**

1. Could this method applicable in non-conversational IR task, e.g. traditional ad-hoc search, if not, what should we change or consider?

2. The method is claimed as proposed for CIR system, while the used datasets Coqa are not retrieval-based. Is it conform to the definition?

3. About the model using, why only use Bert and Bart rather than the dense retriever models?

4. Why choose coqa and clariQ datasets only, rather than the conversational search datasets, such as CAsT, TopiOCQA and QReCC?

**Reasons To Accept:**

1. The problem is interesting and important, which can reduce human labor for the CIR that is considered as the next generation of search engine. The task definition and evaluation methods are clear.

2. The proposed method is valid and promising, and the experimental results are effective and sufficient, showing the better results than existing studies. The chosen tasks are various enough.

3. The paper structure is good and easy to follow. The implementation details are clear and the analysis are sufficient to draw the conclusion.

**Reasons To Reject:**

1. The differences between the proposed Human-Machine Collaboration annotation approach and the traditional human annotated approach is unclear. I suggest to add more description and the statistic information by comparing both, which can obviously view the advantage of  Human-Machine Collaboration annotation approach.

2. The motivation of the method design could be more clear, and explain why it is special for CIR system rather than the traditional IR system.

3. The reason for the chosen datasets and the models are unclear, since they are not directly align to the CIR tasks.

Please also carefully answer the questions for the authors.

**Reproducibility:**

4: Could mostly reproduce the results, but there may be some variation because of sample variance or minor variations in their interpretation of the protocol or method.

**Reviewer Confidence:**

3: Pretty sure, but there's a chance I missed something. Although I have a good feel for this area in general, I did not carefully check the paper's details, e.g., the math, experimental design, or novelty.

---

> ### Author Rebuttal · Authors · 2023-08-28
>
> We thank you for your detailed feedback and suggestions. In the following, we clarify your concerns point by point. Considering some responses and questions are closely related, we answer them together.
>
> # 1.The differences between the proposed Human-Machine Collaboration annotation approach and the traditional human-annotated approach. (R1)
> We appreciate your suggestions and will incorporate additional descriptions in the revised version. As for the annotation process, the traditional approach relies solely on humans to annotate the entire dataset for evaluation. In contrast, our Human-Machine Collaboration approach utilizes both human and machine efforts to collaboratively complete the evaluation task, reducing human labor. Specifically, human labor is reduced by only annotating a carefully selected small subset of the data for evaluation. Machines, in this case, are responsible for selecting which data to annotate and how to calibrate the evaluation results to eliminate biases.
>
> # 2.Motivation for CIR. (R2, Q1)
> We appreciate the reviewer's recognition of the generality of our method. Ideally, our method does have a strong potential for generalization to Information Retrieval (IR) systems. Our motivation for evaluating CIR in this paper lies in two folds.
>
> 1. We consider CIR as a special research area in IR. It is preferable to narrow the scope of our research paper to specific scenarios, such as CIR, in order to conduct a more in-depth analysis and provide a more comprehensive evaluation, rather than exploring general scenarios.
> 2. Compared to IR, CIR evaluation requires more human involvement due to its multi-turn human-machine interactions, making it more labor-intensive for humans. Therefore, it would be more crucial and necessary to ease the human evaluation labor for CIR.
>
> Therefore, we made a successful attempt and the implementation on the CIR setting. Having demonstrated our effectiveness in CIR, we are certainly interested in applying our method to IR in the future :)
>
> # 3.Reason for the chosen datasets and the models. (R3, Q2, Q3, Q4)
> Our definition of CIR aligns with a recent CIR survey[1], which includes conversational QA (i.e., the CoQA dataset) and clarifying questions as common use cases. In these scenarios, We aim to retrieve the answers and questions from the databases, respectively. Particularly on the clarifying question, our experiments include both retrieval-based and generation-based settings, and the results show our effectiveness.
>
> For the sake of convenience in our experiments, we prioritized open-source datasets and models with available checkpoints in these two scenarios. Therefore, based on references [2-4], we used the GODEL model on CoQA for conversational QA evaluation, and BERT, BART, and GPT2 for clarifying question evaluation. To provide a more comprehensive understanding, we conducted additional experiments based on your suggestions. Due to time constraints, we present the results on TopiOCQA [5], where the CIR model is DPR [6] and the surrogate model is FiD [7] (which are used interchangeably). The results demonstrate that our HumCoE approach continues to achieve promising results and outperforms the baselines.
>
> | **CIR** | **#** | **HumCoE (ours)** | **Active Top-K** | **HMCEval** | **MCM** |
> |---------|-------|-------------------|------------------|-------------|---------|
> |   DPR   |     5 |        **82.87%** |           17.05% |    76.19%   |  20.90% |
> |         |    10 |        **77.73%** |           19.43% |  76.19%     |  23.80% |
> |         |    15 |        **82.68%** |           32.58% |      76.19% |  21.30% |
> |         |    20 |        **80.94%** |           24.43% |      76.00% |  23.70% |
> |         |    25 |        **79.81%** |           39.09% |      76.00% |  22.10% |
> |         |    30 |        **79.55%** |           32.58% |      76.00% |  23.60% |
> |   FID   |     5 |        **90.18%** |           30.33% |      72.70% |  15.40% |
> |         |    10 |        **91.98%** |           45.26% |      72.79% |   7.70% |
> |         |    15 |        **90.89%** |           50.28% |      72.78% |   3.80% |
> |         |    20 |        **89.41%** |           65.35% |      72.82% |  10.60% |
> |         |    25 |        **90.62%** |           72.29% |      72.85% |   5.40% |
> |         |    30 |        **90.61%** |           64.11% |      72.83% |   9.20% |
>
> **Reference**
> - [1] Neural Approaches to Conversational Information Retrieval, 2022
> - [2] GODEL: Large-Scale Pre-Training for Goal-Directed Dialog, 2022
> - [3] Building and Evaluating Open-Domain Dialogue Corpora with Clarifying Questions, 2021
> - [4] Towards Facet-Driven Generation of Clarifying Questions for Conversational Search, 2021
> - [5] TopiOCQA: Open-domain Conversational Question Answering with Topic Switching, 2022
> - [6] Dense Passage Retrieval for Open-Domain Question Answering, 2020
> - [7] Leveraging Passage Retrieval with Generative Models for Open Domain Question Answering, 2021

---

### Meta-Review · Area_Chair_cixF · 2023-09-19

**Recommendation:** 4

**Metareview:**

The paper presents a novel approach to address the problem of automatic evaluation for conversational information retrieval (CIR) tasks. The authors propose a method called HumCoE that leverages active testing to reduce the amount of human labor required for evaluation. The selection of representative data for human annotation is done using a surrogate model, which generates pseudo labels that differ from the CIR model's prediction. The paper presents empirical results, demonstrating that HumCoE can produce evaluations comparable to full human evaluation with much less human labor. The authors apply this method across multiple CIR tasks, showing its versatility and effectiveness.

The paper addresses a significant challenge in CIR research, particularly the labor-intensive aspect of human evaluation. The proposed method brings together active learning and human-machine collaboration, making it an innovative solution in the field. The paper presents a comprehensive empirical analysis across multiple datasets and tasks, demonstrating the effectiveness of the approach compared to baseline methods. The application of active testing in CIR is a novel contribution, and the approach could potentially inspire future research in human-machine collaboration for system evaluation.

While the paper's contributions are notable, there are several areas where it could be improved. One of the main concerns raised by the reviewers relates to the clarity and depth of explanation in the methods section, which seemed to be hard to understand. The use of the surrogate model could be better explained, and it is suggested that a few running examples could help clarify this. There are also questions about the sensitivity of the final HumCoE scores to the choice of the surrogate model. It would be beneficial to include more discussion or experiments showing how using different models for the surrogate model affects the scores. It has also been highlighted the need for a clearer explanation of the evaluation metrics used (or cite relevant works), namely the stability and consistency ones, which will increase the readability. It would be helpful if the authors could provide references to other works that use these metrics, or explain them in more depth. The choice of datasets and models used in the study has also been questioned. The reviewers have asked for more clarity on why certain datasets were chosen, and why only Bert and Bart were used instead of other models. They have also suggested that the authors could provide more detail on scenarios where HumCoE may not work as well, perhaps through additional error analysis.

Despite these areas for improvement, the paper presents an innovative approach to a significant problem in CIR research. The authors are encouraged to address the concerns raised by the reviewers to further strengthen the paper. In particular, the authors should provide more clarification on their methodology, the choice of datasets and models, and the evaluation metrics used. Further experimentation with different surrogate models and a deeper analysis of scenarios where HumCoE may not work well could also enhance the paper. AC acknowledged that the authors tried to address most of these issues during rebuttal.

---

### Decision · Program_Chairs · 2023-10-07

**Decision:**

Accept-Main

**Comment:**

The paper presents a novel approach to address the problem of automatic evaluation for conversational information retrieval (CIR) tasks. The authors propose a method called HumCoE that leverages active testing to reduce the amount of human labor required for evaluation. The selection of representative data for human annotation is done using a surrogate model, which generates pseudo labels that differ from the CIR model's prediction. The paper presents empirical results, demonstrating that HumCoE can produce evaluations comparable to full human evaluation with much less human labor. The authors apply this method across multiple CIR tasks, showing its versatility and effectiveness.

The paper addresses a significant challenge in CIR research, particularly the labor-intensive aspect of human evaluation. The proposed method brings together active learning and human-machine collaboration, making it an innovative solution in the field. The paper presents a comprehensive empirical analysis across multiple datasets and tasks, demonstrating the effectiveness of the approach compared to baseline methods. The application of active testing in CIR is a novel contribution, and the approach could potentially inspire future research in human-machine collaboration for system evaluation.

While the paper's contributions are notable, there are several areas where it could be improved. One of the main concerns raised by the reviewers relates to the clarity and depth of explanation in the methods section, which seemed to be hard to understand. The use of the surrogate model could be better explained, and it is suggested that a few running examples could help clarify this. There are also questions about the sensitivity of the final HumCoE scores to the choice of the surrogate model. It would be beneficial to include more discussion or experiments showing how using different models for the surrogate model affects the scores. It has also been highlighted the need for a clearer explanation of the evaluation metrics used (or cite relevant works), namely the stability and consistency ones, which will increase the readability. It would be helpful if the authors could provide references to other works that use these metrics, or explain them in more depth. The choice of datasets and models used in the study has also been questioned. The reviewers have asked for more clarity on why certain datasets were chosen, and why only Bert and Bart were used instead of other models. They have also suggested that the authors could provide more detail on scenarios where HumCoE may not work as well, perhaps through additional error analysis.

Despite these areas for improvement, the paper presents an innovative approach to a significant problem in CIR research. The authors are encouraged to address the concerns raised by the reviewers to further strengthen the paper. In particular, the authors should provide more clarification on their methodology, the choice of datasets and models, and the evaluation metrics used. Further experimentation with different surrogate models and a deeper analysis of scenarios where HumCoE may not work well could also enhance the paper. AC acknowledged that the authors tried to address most of these issues during rebuttal.